# Effects of a Low-Volume Kettlebell Strength Program on Vertical Jump and Isometric Strength in Dancers: A Pilot Study

**DOI:** 10.3390/jfmk10020130

**Published:** 2025-04-11

**Authors:** Yaiza Taboada-Iglesias, Iria Filgueira-Loureiro, Xoana Reguera-López-de-la-Osa, Águeda Gutiérrez-Sánchez

**Affiliations:** 1Special Didactics Department, Faculty of Science Education and Sport, University of Vigo, 36005 Pontevedra, Spain; xreguera@uvigo.gal (X.R.-L.-d.-l.-O.); agyra@uvigo.gal (Á.G.-S.); 2Education, Physical Activity and Health Research Group (Gies10-DE3), Galicia Sur Health Research Institute (IIS), University of Vigo, 36005 Pontevedra, Spain; 3Physical Activity and Sport Sciences, University of Vigo, 36005 Pontevedra, Spain; iria.fl@hotmail.com

**Keywords:** dancers, *dehors*, kettlebell, strength training, vertical jump, isometric strength

## Abstract

**Objectives**: The jumping ability and strength of the lower limbs in dancers are fundamental to their artistic performance. Similarly, the correct placement of the various leg positions, such as parallel and *dehors*, are essential aspects of dance technique. We aimed to analyze the effectiveness of a modified strength program on jumping capacity in parallel and *dehors*, and to assess whether this type of training improves the isometric strength of dancers. **Methods**: An experimental research study was conducted with pre- and post-intervention assessments on a group of nine semi-professional dancers (seven women and two men) (X = 20 years and DT = 6.67), with an average weight of 62.12 ± 7.38 kg and a height of 1.67 ± 0.05 m. Body composition, isometric strength in the lower limbs, and vertical jump capacity with feet in parallel and in the *dehors* position were analyzed. The “Simple and Sinister” kettlebell training protocol was implemented, reducing the number of sessions and the duration of the program while incorporating a progression in load. **Results**: Significant changes were observed in both the parallel jump and the *dehors* position; however, body composition and isometric strength, although showing higher values at the end of the intervention for all variables, did not demonstrate significant improvements. Thus, while in the pre-intervention phase, jump capacity was associated with multiple variables, in the post-intervention phase, an inverse association was found only with the percentage of body fat. **Conclusions**: A 12-week training program with two sessions per week is sufficient to achieve significant changes in the jumping ability of dancers, but not in the isometric strength of the lower limbs.

## 1. Introduction

Dance originated in prehistory as a means of representation and social and religious communication. It is an artistic expression used by humans to spontaneously convey their emotions and feelings [1,2,3,4].

The foundation of contemporary dance is found in classical ballet, which originated from an Italian dance that was codified in 1661 by the Royal Academy of Music and Dance at the court of Louis XIV. This Academy established a code of steps, placements, and movements, thereby creating the basic positions and defining specific steps used in the world of dance. One of the basic positions is the *dehors*, which aims to rotate each leg by 90°, forming a 180° angle between both feet [5,6]. This position in dancers exhibits a strength deficit for the execution of vertical jumps compared to the position with feet in parallel, as the muscular effort required to maintain the position hinders the activation of the muscles involved in jumping [5].

One of the essential physical capacities in dance is strength, which is necessary for the jumps, turns, and balances that are performed constantly in this discipline [5,7]. Despite this, specific strength training has been rejected by academies for many years [8,9]. One of the main reasons for this rejection is the failure to consider strength training as necessary in the world of dance, despite evidence that load training, when tailored to dancers, can enhance muscular power, thereby achieving better results in explosive movements [5,10]. There is also a fear of excessive hypertrophy that could compromise the aesthetic appearance of dancers [11]. In addition, dancers’ practice schedules are often quite tight and can lead to discarding workouts that they do not consider a priority, as perfecting their art demands long hours [12] and requires high levels of physical fitness and artistic ability [13], so strength training is a practice that is not widely adopted in dance schools or conservatories.

Regarding strength training in dancers, the importance of specific strength work seems clear for them to perform movements with correct technique and precision [5,7,14,15]. It has been demonstrated that isometric strength training helps improve stability in static positions and contributes to the execution of movements without losing balance. Additionally, strength training not only improves the quality of movements but also helps prevent injuries [15,16].

Strength training aims to improve the recruitment of motor units, especially fast motor units, and enhance intermuscular coordination [17]. This helps improve explosive movements, including vertical jumps [18]. These adaptations can be achieved through Tsatsouline’s kettlebell training method [19], “Simple and Sinister”, which involves the execution of the one-hand swing exercise and the Turkish get-up exercise three times a week for five months. Kettlebell training is considered more ergonomic compared to other equipment like barbells or dumbbells, and it may be more fitting for delivering ballistic movement. Therefore, the exercises are performed with greater safety, especially for individuals with little experience in load-bearing work. This program involves using the same weights from the beginning. For this reason, it seems to be necessary to implement progressive loading based on the principle of progression, which helps prevent injuries. In this regard, previous studies on beginner kettlebell training recommend starting with an eight-kilogram weight for women and a twelve-kilogram weight for men, increasing the load progressively by 4 kg [20].

This type of training has been shown to produce physiological changes in the body and improve the vertical jump ability and postural control [7,19]. In the original protocol by Grigoletto [19], a weekly training frequency of three sessions was used. However, other studies demonstrated positive results with a training frequency of two times per week [20]. In fact, studies investigating untrained subjects found no significant differences between training three times a week and once a week [21]. This is interesting in terms of facilitating the inclusion of programs that are shorter, more practical, and convenient to adopt in populations with such tight schedules as dancers.

For these reasons, strength training is necessary and beneficial for dancers, and it is crucial that dance academies implement effective and engaging programs that do not require a significant time commitment, facilitating their integration into the dancers’ daily routines. The aim of this work is to analyze the effectiveness of a modified strength program in improving jumping ability in both parallel and *dehors* positions. For this research, the number of sessions and the duration of the original program were reduced, and the load progression was adjusted, starting with lower weights to minimize the risk of injury. Furthermore, it seeks to determine whether this type of training contributes to improving dancers’ isometric strength. The hypothesis of this research is that the “Simple and Sinister” kettlebell training program, using progressive loading, can improve lower-limb strength and jumping ability in a shorter time in both parallel and *dehors* positions.

## 2. Materials and Methods

### 2.1. Participants

An experimental pre- and post-intervention study was conducted on a sample of 9 dancers (7 women and 2 men) with an average age of 20 (SD = 6.67), an average weight of 62.12 ± 7.38 kg, and an average height of 1.67 ± 0.05 m. The participants were semi-professional dancers of a teaching academy receiving more than 7 h of weekly training in various types of dances with a mean of 10 h (range 7–18 h) per week, including at least one ballet class. The participants had an average experience of 6.9 years (range 3–10 years), of which between 2 and 4 years were spent on professional training. Dancers with serious injuries, surgeries of the lower limbs or spine, or physical limitations preventing load-bearing exercises were excluded.

### 2.2. Ethics

This study adhered to the ethical procedures outlined by the Organic Law on Personal Data Protection (Organic Law 15/1999 of 13 December 1999). All procedures conducted in this research were approved by the Ethics Committee of the Faculty of Education and Sport Sciences at the University of Vigo (reference number 06-220323, approval date 22 March 2024) and followed the agreements of the Helsinki Declaration. All participants signed an informed consent form, and for minors, the form was signed by their legal representatives. Participants were informed of the purpose of the study and the procedures to be followed. The data treatment was coded to ensure anonymity, and participation was voluntary.

### 2.3. Procedure

#### 2.3.1. Body Composition Measurement

First, a body composition study was performed using a bioimpedance scale (Tanita bc-601) to assess weight, body fat percentage, and muscle mass.

#### 2.3.2. Dynamometry

Next, isometric strength was measured using a “MicroFET 2” (Hoggan Health Industries, Salt Lake City, UT, USA) dynamometer to evaluate hip flexors, extensors, external rotators, and knee flexors and extensors using the following protocols (Figure 1):Isometric Strength of Hip Flexors (ISHF): In a supine position with the leg bent at 90° of hip flexion, the device was placed 5 cm above the proximal edge of the patella. The participants were instructed to apply pressure against the device for 5 s against maximum resistance [22].Isometric Strength of Hip Extensors (ISHE): In a prone position with the leg bent at 90°, the device was placed 5 cm above the knee joint line on the back of the thigh. The participants applied pressure against the device for 5 s against maximum resistance [22].Isometric Strength of Knee Flexors (ISKF): In a prone position with the knee straight, the device was placed on the posterior region of the heel. The participants applied pressure against the device for 5 s against maximum resistance [23].Isometric Strength of Knee Extensors (ISKE): Seated with the legs perpendicular to the floor and knees bent at 90°, the device was placed on the anterior leg 5 cm above the perpendicular line with the ankle. The participants applied pressure against the device for 5 s against maximum resistance [24].Isometric Strength of External Hip Rotators (ISHER): In a prone position with the knee bent at 90°, the device was placed 5 cm above the tibial malleolus, and the participants applied maximum isometric contraction against the device for 5 s [22].

Three repetitions were performed on each leg, leaving 30 s of rest between repetitions, and the repetition of maximum force output was selected for further analysis. All measurements were performed by the same investigator, who was previously trained in the measurement protocols. The intervention was started after the completion of the training examinations. The pretest was conducted in a transitional stage and the intervention during the second special preparatory period, prior to the final performances. The posttest was conducted 24 h after the last intervention session.

**Figure 1 jfmk-10-00130-f001:**
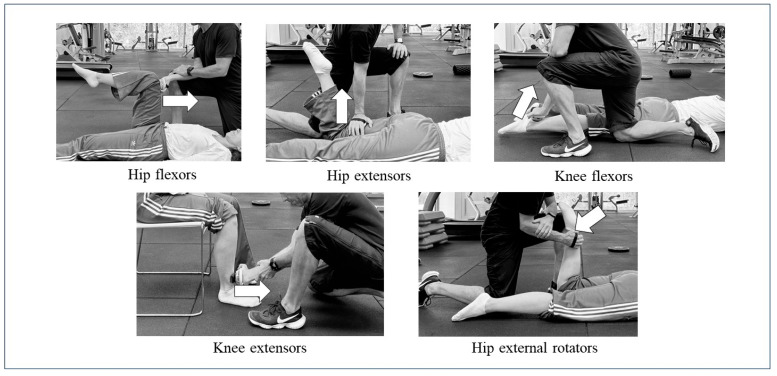
Isometric strength measurement protocol of the analyzed lower limb musculature.

#### 2.3.3. Vertical Jump Capacity

Finally, explosive lower limb strength was measured using vertical jump tests. The “Counter Movement Jump” from Bosco’s test was used, which starts from a standing position with extended knees, followed by a rapid flexion–extension of the knees, and then a maximum vertical jump [25]. The jump was analyzed in both parallel and first *dehors* position (Figure 2). The jumps were recorded using an iPad at 60 fps and analyzed using the My Jump Lab (My Jump 3) (My Jump Technologies S.L., Madrid, Spain). app for iOS [26]. An average of 3 attempts in each position was taken for analysis.

### 2.4. Intervention Protocol

The participants continued with their regular dance classes during the 12 weeks of the intervention, incorporating two 15 min sessions per week following the “Simple and Sinister” program [27]. This modified program significantly reduced the volume of the original intervention. It consists of a simple dynamic warm-up followed by 100 kettlebell swings (50 with each hand) (Figure 3) and 10 Turkish get-ups (5 with each hand) (Figure 4).

The swing exercise involves a ballistic movement with one arm, swinging the kettlebell between the legs with the arm extended, focusing the effort on the hips and legs. The Turkish get-up is performed slowly, starting from a lying position while holding the kettlebell above the head with an extended arm. The dancer then sits up with the help of the free arm, flexes the leg on the same side as the kettlebell, and raises the hips. From this position, the outstretched leg is placed under the body to kneel, and finally, the dancer stands up, before reversing the process to return to the floor.

The first exercise (swings) was performed in 5 min, with 10 repetitions every 30 s, and the second exercise (Turkish get-ups) in 10 min, with one repetition per minute.

Initially, the following protocol [28] was followed to determine the kettlebell weights: 8 kg for women and 12 kg for men. In each training session, the subjective effort scale (Borg CR-10) was recorded for each exercise [29]. The weight was increased by 4 kg when scores of 4 or lower were recorded in the previous session.

### 2.5. Data Analyses

Normality was assessed using the Shapiro–Wilk test. The Greenhouse–Geisser adjustment was applied when Mauchly’s sphericity test was violated (*p* < 0.05 for Mauchly’s sphericity test) in the case of a related sample analysis (pretest/posttest). The mean (X) was used as the central tendency measure, and the standard deviation (SD) as the dispersion measure for descriptive analysis in both the pretest and posttest. Changes in dependent variables between pretest and posttest were compared using a paired *t*-test. To analyze the influence of anthropometric and isometric strength variables on jumping ability both in parallel and *dehors*, we proceeded to perform a Generalized Linear Mixed Model (GLMM). We included a fixed effect of timing (pretest and posttests) and the identification of each subject as a random effect to control interindividual variability. A compound symmetry covariance structure was selected as the best fit of the model and the coefficients were estimated using the restricted maximum likelihood method. Statistical significance was set at *p* < 0.05, and when appropriate, Cohen’s d was used to determine the effect size, where values of 0.2, 0.5, and 0.8 represent small, medium, and large differences, respectively. Statistical analysis was performed using SPSS version 21.0 statistical software.

## 3. Results

The participants completed the training program with 100% adherence. The data from the studied variables followed a normal distribution. The progression of the load and the perception of effort achieved in each training session are shown in Figure 5. It can be seen that five participants had to increase the weight and that all participants experienced a decrease in the perception of effort throughout the program.

Table 1 presents the descriptive analysis of the variables analyzed before and after the intervention. Significant changes were found in both the parallel and *dehors* jump positions (*p* = 0.003 and *p* = 0.000, respectively), demonstrating greater jump capacity in both positions after the intervention. However, while body composition and isometric strength values improved across all variables, these improvements were not statistically significant. The reliability of the jump and isometric strength measurements was assessed using the intraclass correlation coefficient (ICC) and the coefficient of variation (CV). The ICC results indicate excellent reliability for all measurement variables. In addition, the CV values are low, suggesting minimal variability between measurements within each subject, which reinforces the stability and consistency of the data obtained.

Cohen’s d for the vertical jump achieved a large effect size. For the *dehors* jump, Cohen’s d was 1.08 (r = 0.48), also indicating a large effect size.

Associations between anthropometric variables and isometric strength and jump performance were explored.

The resulting linear mixed models were significant in both types of jumps (parallel: F = 4,874,751.464; *p* < 0.001, and in *dehors*: F = 3,961,927.249; *p* < 0.001), which shows that jumping capacity is largely explained by the variables included in the model.

In both jumps, timing was observed as a significant effect (parallel: F = 47.471; *p* = 0.020, and in *dehors*: F = 548.323; *p* = 0.002), which reaffirms that the training program influenced performance.

Analyzing the model in the parallel jump, none of the variables showed a significant association with the increase in jumping. However, in the *dehors* jump, fat percentage (F = 38.764; *p* = 0.025; B = −4.103), muscle mass (F = 40.494; *p* = 0.024; B = −6.303), left hip extension strength (F = 117.944; *p* = 0.008; B = −0.069), right knee flexor strength (F = 84.103; *p* = 0.012; B = 0.098), right knee extensor strength (F = 37.014; *p* = 0.026; B = −0.039), right hip external rotator strength (F = 58.912; *p* = 0.017; B = 0.112), body weight (F = 34.089; *p* = 0.028; B = 4.327) and BMI (F = 29.083; *p* = 0.033; B = −1.910) were found to be significant predictors of jumping ability. Figure 6 shows the directions of the estimation coefficient of all the variables included in the model. It shows that depending on the time point, the jump was greater in the posttest than in the pretest, demonstrating the improvement after the program. The results emphasize that in both types of jumping, the higher the BMI and fat mass, the lower the jumping capacity; however, the higher the body weight, the greater the jumping capacity, but the higher the muscle mass, the lower the jumping capacity. In general terms, the isometric strength variables are positively related, indicating that the greater the strength, the greater the jump. Among the four variables referring to isometric strength that were significant in the model in the external rotation jump, two of them obtained a positive relationship with jumping (right knee flexion and right hip external rotators); however, greater strengths of the left hip extensors and right knee extensors appear to be related to a lower jumping capacity.

## 4. Discussion

The aim of this study was to evaluate the effectiveness of a modified strength program in improving jump ability in semi-professional dancers. The key findings were as follows: (1) the exercise protocol does improve jump ability in both parallel and *dehors* positions; (2) two weekly sessions are sufficient to achieve these improvements; (3) it is not necessary to use the high weights prescribed in the original protocol; improvements can be made starting with lighter weights and controlling progression using the Borg scale; (4) significant improvements in isometric strength were not observed; (5) The variables of isometric strength or body composition do not seem to have the power to explain the improvement in the parallel jump, but they do in the *dehors* jump.

A previous study demonstrated that improvements in jump ability in both positions could be achieved with a five-month training program [19]. In this study, similar results were achieved in just three months. Thus, it seems that the “Simple and Sinister” program carried out in only two weekly sessions over a period of three months already showed improvements in the dancers’ jumping ability in both parallel and *dehors* positions.

This study utilized progressive loading, even though the original training program employed fixed loads based on gender, as used in the study by Grigoletto et al. [19]. This progression of loads was based on the study by Jay et al. [28], which recommended starting with 8 kg for women and 12 kg for men, with potential weight increases as the intervention progressed. In this study, a Borg scale feedback system was introduced, where participants rated their perceived exertion at the end of each session, providing feedback on when to increase the weight. The participants’ perception of effort decreased throughout the program, with five of them having to increase their workload during the intervention. This progressive loading approach likely contributed to the adherence to the study, as the challenge of increasing weights served as an additional source of motivation for the dancers. Since the study participants, though athletes, had no prior experience with load training, the use of progressive kettlebell training was ideal, as it has been shown in previous studies to reduce lower-back, neck, and shoulder pain [28]. In addition, for four of the participants, the initial load of 8 kg for women and 12 kg for men was sufficient, as proposed by Jay et al. [28], which suggests that for this population the minimum “simple” target weights proposed in the original program (16 kg for women and 32 kg for men) [30] are too high.

Regarding the results on body composition and isometric strength, no significant differences were found after the implementation of the program. A study by Ferragut et al. [31] demonstrated that maximum isometric strength is not a reliable predictor of vertical jump height, indicating that the two variables are not directly related, which can be seen in our study with the parallel jump according to the linear mixed model. However, this study shows that while isometric strength did not significantly change, vertical jump performance did. Although the isometric strength variables are not individually determinant in predicting improvement in parallel jumping, some of them do seem to affect the ability to jump in *dehors*. Increased strengths of the right hip external rotators and right knee flexors seem to predict an improvement in *dehors* jumping. However, increased jumping in this position is also related to low strength values of the left hip extensors and right knee extensors, despite previous studies showing that they are a key part of the propulsion in vertical jumping [32]. These results may be because in this position, this musculature cannot apply force equally, the rotator musculature becoming more important due to the need to maintain the position in external rotation.

Strength programs that induce greater isometric strength are those that use high loads [33]. In the training of the lower-limb musculature, it has been proven that high loads (80% 1RM) are the ones that obtain the greatest improvements in isometric strength, although low loads (30% 1RM) can also obtain positive results [34]. However, others indicate that only high loads of 80% of 1RM, and not lower loads around 20–40% of 1RM, are the ones that cause improvements in isometric strength [35]. This may be the key factor explaining the insignificant isometric strength improvement shown in our study, since we used lower loads than those set by the original program. Despite this, in untrained young people, static training provokes improvements in isometric strength at both 80% and 50% of the maximum voluntary contraction [36], which could be interesting for dance practitioners due to its specificity, making it possible to train with slightly lower loads.

In our study, we cannot confirm that a high muscle mass percentage is related to higher jumps, as considered by Ferragut et al. [31], who confirmed a direct relationship between lower limb muscle mass and vertical jump height. While muscle mass improved across the entire body in this study, the improvements were not statistically significant. In addition, the coefficients estimated in the mixed linear model indicate, even though it is not a significant variable in the model for the parallel jump, that the relationship is indirect, indicating that the greater the muscle mass, the lower the jump both in parallel and in *dehors*. Therefore, it seems that in this type of jumping, it is determinant to focus more on neuromuscular control and muscle synergies, rather than on the gain of muscle mass. Despite this, the results also show an inverse relationship with fat mass [37]. However, BMI was also inversely related, and weight was directly related, showing that greater weights were related to greater jumping capacity, which could be related to greater height. Despite this, when the linear model was run with height, it did not appear significant and generated less explanatory power in the model. This could indicate that other variables such as leg or thigh length could be affecting the biomechanics of jumping, inducing a greater use of elastic energy, which seems to be key, especially in the *dehors* position. Further studies in this regard are warranted to confirm the relationship between these anthropometric variables and *dehors* jumping performance.

One of the main limitations of this study is the small sample size. With only nine dancers, the results cannot be generalized. Another important limitation of this study is the absence of a control group that would allow more robust results to be established. However, given the novelty of the subject under study and the difficulty of obtaining a large and homogeneous sample, we opted for the pilot design presented in this study. Additionally, the study is limited to a young population with no prior strength training experience. Future research should increase the sample and consider a control group design. Studies could also be considered in which low-volume strength programs could be implemented to also improve isometric strength, since this is an indispensable capacity in dance, and to study the relationship with the improvement of the vertical jump in the *dehors* position.

This study suggests that the “Simple and Sinister” training method [27], performed twice a week, can be applied effectively, as it has been shown to improve jump performance in dancers. Given the 100% adherence rate among participants, this method does not present an excessive burden. Additionally, since the program takes only 15 min per session, it does not demand too much time and could be integrated into dancers’ regular training sessions.

## 5. Conclusions

A 12-week plan with just two sessions per week was sufficient to produce significant improvements in dancers’ jump ability, but not in lower limb isometric strength. It is not necessary to start with heavy weights; instead, a progressive approach, with effort controlled through the Borg scale, is effective for this population.

Moreover, the low load strength training program using kettlebells did not significantly increase the muscle mass of our dancers. Therefore, dance teachers and dancers should consider adopting this training method to enhance the dance performance without the significant observable morphological changes (e.g., bulky muscle mass).

## Figures and Tables

**Figure 2 jfmk-10-00130-f002:**
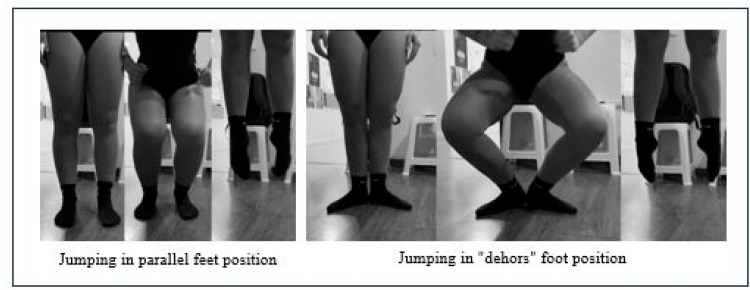
Parallel and first “*dehors*” position jumps.

**Figure 3 jfmk-10-00130-f003:**
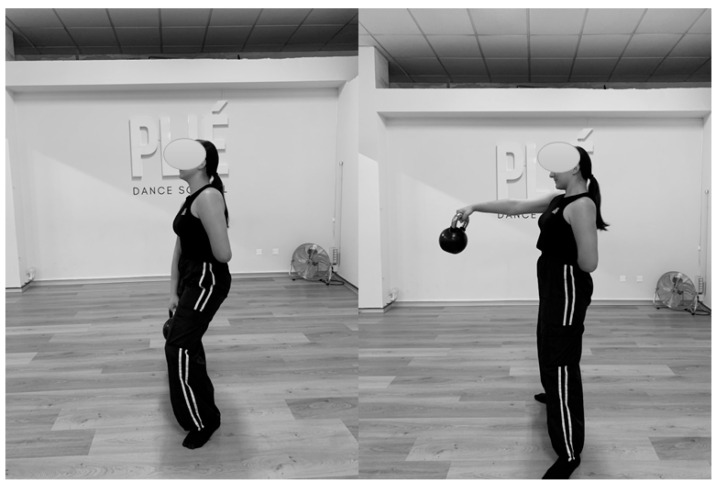
One-hand swing exercise.

**Figure 4 jfmk-10-00130-f004:**
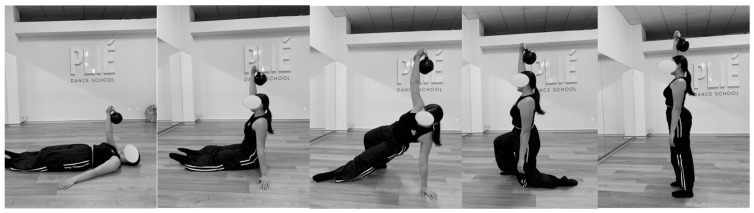
Turkish get-up exercise.

**Figure 5 jfmk-10-00130-f005:**
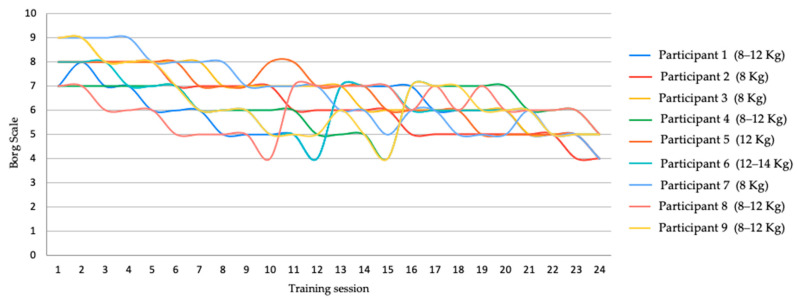
Progression of the load and perception of effort by Borg Scale of each participant.

**Figure 6 jfmk-10-00130-f006:**
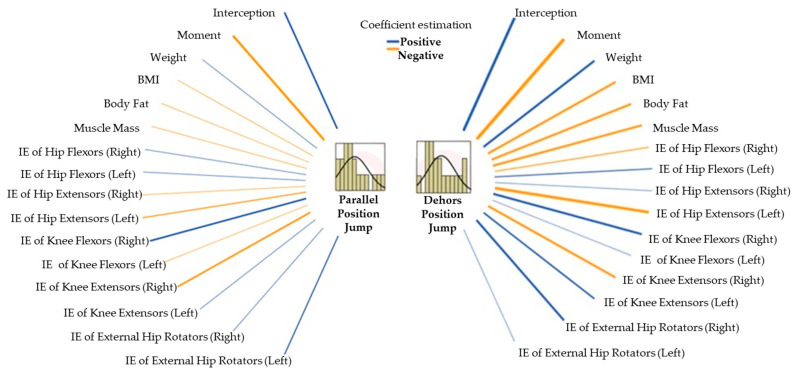
Estimated coefficients from the generalized linear mixed model for jump performance (*the line width shows the magnitude of the coefficient estimations*).

**Table 1 jfmk-10-00130-t001:** Descriptive statistics (means and standard deviations) of the analyzed variables before and after the intervention, and comparisons between pretest and posttest.

			Pretest (n = 9)			Posttest (n = 9)			
			X (SD)	ICC (95%)	Mean CV (%)	X (SD)	ICC (95%)	Mean CV (%)	*p*-Value
Weight (kg)			62.12 (7.38)			62.71 (7.37)			0.257
Body Mass Index (kg/m^2^)			22.29 (2.62)			22.73 (2.51)			0.148
Body Fat %			22.39 (6.46)			21.48 (7.94)			0.197
Muscle Mass (kg)			45.69 (6.16)			46.66 (6.40)			0.053
Isometric Strength	Hip Flexors	(R)	171.36 (44.16)	0.901	8.7	178.98 (26.65)	0.953	2.6	0.541
		(L)	169.12 (38.10)	0.927	4.8	185.92 (27.26)	0.882	4.6	0.073
	Hip Extensors	(R)	199.22 (53.17)	0.935	5.8	232.02 (86.88)	0.979	4.1	0.125
		(L)	196.49 (60.46)	0.896	7.8	209.42 (70.03)	0.963	5.2	0.518
	Knee Flexors	(R)	204.32 (48.94)	0.886	5.4	211.86 (40.09)	0.894	3.6	0.186
		(L)	198.73 (50.52)	0.917	6.6	210.01 (38.83)	0.975	2.4	0.444
	Knee Extensors	(R)	247.46 (66.52)	0.938	6.0	249.80 (53.26)	0.934	4.9	0.902
		(L)	243.41 (43.70)	0.869	5.6	258.40 (44.43)	0.947	3.8	0.435
	External Hip Rotators	(R)	117.24 (19.67)	0.881	4.9	122.31 (25.85)	0.955	3.0	0.266
		(L)	132.86 (31.46)	0.902	5.1	122.81 (25.89)	0.934	4.1	0.288
Parallel Position Jump (cm)			25.31 (5.41)	0.948	3.7	29.91 (4.93)	0.915	3.5	0.003 *
*Dehors* Position Jump (cm)			21.60 (4.64)	0.890	4.9	26.54 (4.48)	0.835	5.9	0.000 *

Note: * *p* < 0.05 indicates statistical significance. L: left; R: right.

## Data Availability

Data may be requested from the corresponding author.

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
