# Peer review of "Effects of a Low-Volume Kettlebell Strength Program on Vertical Jump and Isometric Strength in Dancers: A Pilot Study"

_jfmk, 2025, doi:10.3390/jfmk10020130_

Round 1

Reviewer 1 Report

Comments and Suggestions for Authors

Thank you for your submission. I have several major concerns from this investigation that would need to be addressed before recommending for acceptance.

My first concern is in regard to the isometric strength testing. What controls were put in place to ensure that accurate and valid data was obtained? The handheld dynamometer introduces a significant amount of error into the data.

The second concern I have involves the increase in jump height but no changes in strength. Jump performance can be narrowed down to strength to mass ratios. As strength did not change and body mass did not change what explanation do you have for the improvement in the jump height?

Line 258: There is nearly identical values for many of these metrics, you do not have data supporting this statement.

Overall the discussion needs to have much more detailed information of the lack of changes that may have been seen.

Author Response

POINT-BY-POINT FOR THE REVIEWERS

The authors would like to thank all the reviewers for their comments and suggestions, and for their invaluable help in substantially improving our work. We will now proceed to answer each of the questions requested by the reviewers. 

Reviewer 1:

Dear reviewer, we are very grateful for your comments and we will try to resolve and respond to each of them. In the text of the document we will provide the changes highlighted in red. 

Comments to the author

Thank you for your submission. I have several major concerns from this investigation that would need to be addressed before recommending for acceptance.

My first concern is in regard to the isometric strength testing. What controls were put in place to ensure that accurate and valid data was obtained? The handheld dynamometer introduces a significant amount of error into the data.

We understand your concern, we ourselves had it in the design of the study, however, manual dynamometry provided us with a very high applicability as we could move around with the instrument without problems and could easily adapt it to different measurements. In addition, it is a validated measurement for the tests performed (Thorborg et al. 2013; Kelln et al. 2008).

  1. Thorborg, K.; Petersen, J.; Magnusson, S.P.; Hölmich, P. Clinical assessment of hip strength using a hand‐held dynamometer is reliable. J. Med. Sci. Sports. 2010, 20 (3), 493–501. https://doi.org/10.1111/j.1600-0838.2009.00958.x
  2. Kelln, B.M.; McKeon, P.O.; Gontkof, L.M.; Hertel, J. Hand-Held Dynamometry: Reliability of Lower Extremity Muscle Testing in Healthy, Physically Active, Young Adults. Sport Rehabil. 2008, 17 (2), 160–170. https://doi.org/10.1123/jsr.17.2.160

Likewise, to ensure that the precision of the measurements was carried out using the protocols shown as valid evidence for the measurement of these muscle groups, as can be seen in the section on material and methods. These protocols indicate the exact point (from anatomical points) and the specific anatomical range for each measurement, which allows for more precise measurements, from a biomechanical point of view.

In addition, the measurements were always performed by the same evaluator, previously trained in the protocols. In this way we avoided variability between measurements. We have added this point in the material and methods section (line 140). 

The second concern I have involves the increase in jump height but no changes in strength. Jump performance can be narrowed down to strength to mass ratios. As strength did not change and body mass did not change what explanation do you have for the improvement in the jump height?

It is a very interesting question that you raise, however, it should be noted that jumping capacity did improve and therefore explosive strength was increased, while isometric strength did not improve. This may be because the improvement was due to better neuromuscular coordination between muscle groups and increased recruitment of motor units typical of dynamic contractions, as well as myotatic reflex efficiency may have improved. However, isometric strength does not depend on the stretch-shortening cycle or the speed of contraction, but rather on total motor unit recruitment and sustained contraction, so it seems to indicate that the improvement in strength with a Kettlebell program is due to these types of factors.

Line 258: There is nearly identical values for many of these metrics, you do not have data supporting this statement.

We have reworded the paragraph so that the data obtained support the statements made. Thank you very much for having detected the error.  

Overall the discussion needs to have much more detailed information of the lack of changes that may have been seen.

The discussion has been reformulated to a large extent, we hope that we have been able to respond sufficiently to your requirements.

Reviewer 2 Report

Comments and Suggestions for Authors

Thanks for the effort and the potential contribution in the dance areas. The major weaknesses of this study include 1) lack of control group; 2) small sample size. Theoretically, most submission will be rejected for interventional study with no control group for comparison. Please if possible provide justification why this study design was performed. The following are more specific comments for your considerations:

Introduction

Line 47 - 54: not sure if you can find a bit evidence that, to be honest, many pre-professional, semi-professional and professional dancers are required to practice with a very tight schedule. Additional time and efforts on strength training are not their priority........I am thinking if this point is also useful that the training equipment and program must be therefore handy, versatile and convenient to adopt

Line 81 - 84 a bit strange to mention "led by a professional in physical activity and sports"......"design effective and engaging programs...."not sure what is the implication if you study is not to investigate the exercise instruction skill things

Line 55-67: Kettlebell may be more capable in delivering ballistic movement, the release of power and follow-through................just for your consideration if you find this also useful to be added in somewhere

Line 74-80: I am thinking if this paragraph can be better relocated and also the line 75-76, it is not quite matching the key content of this paragraph

Methods:

Line 100 - 7 hours per week only?

Figure 1 - ISKF at 90 degree? but the picture is not 90. Also I wonder if the researcher in the picture can hold the leg securely for measuring the powerful hip flexors and knee extensors without a good body position / mechanics

Any verbal encouragement for isometric test?

Line 139 - what is the meaning of choosing the maximum repetition?

Line 156 - any warm up required such as 11+ Dance or simple dynamic warm up?

Line 188-189 - Why Pearson's correlation used....what is the purpose to identify the association?

Results:

Table 1 is not presented in a robust manner. It should have Pretest (mean ± SD), Posttest (mean ±SD), Differences (Post-Pre), p-value, Cohen's d value in one table. Also Isometric Strength as a section --> Hip flexors as a sub-section then L and R as two rows......now the left hand side so many words and difficult to read

Line 208-228: I don't see any big value of reporting all the correlation. If you want to show any change of isolated muscle strength affecting the jumping performance, please consider using a multiple linear regression OR plot a heatmap with corrplot with different colors to help visualize their relationship. Now all these findings are in the long text with poor readability.

Discussion:

Line 230-241: this paragraph can be more precise. It is mainly to highlight your findings or research purpose but now a bit long and clumsy

Line 244: I don't think you can confirm this if you have no control group in your study

Line 251: where is the Borg RPE results? Can you plot a line chart showing the trend of 12 weeks and also the KB weight used as the barchart? Or if this chart is not the best, please if possible also show the result of this part. You mentioned about progressive but we can't fully recognize how progressive load to be implemented in your study

Line 258-267: may consider citing some previous studies regarding the required load and dosage to improve the maximum isometric strength. And now your program due to the light load in nature, it may not reach the threshold to induce sufficient intramuscular coordination

Line 268-272: Please CAREFULLY justify why no control group is used in your study. It is the biggest limitation even more severe than the small sample size. Maybe it is a pilot study using such a novel concept in dancers? Maybe the number of semi-professional dancers or professional dancers are always lacking? Anyway, you still need to point this out and also suggest future study to include the control group or even any other interesting group such as fixed load group such that Control vs Progressive vs Fixed ?

Comments on the Quality of English Language

Grammar and sentence structure are good overall. Just some content arrangements can be better

Author Response

POINT-BY-POINT FOR THE REVIEWERS

The authors would like to thank all the reviewers for their comments and suggestions, and for their invaluable help in substantially improving our work. We will now proceed to answer each of the questions requested by the reviewers. 

Reviewer 2:

Dear reviewer, we are very grateful for your comments and we will try to resolve and respond to each of them. In the manuscript  we will provide the changes highlighted in red.

 Comments to the author

Thanks for the effort and the potential contribution in the dance areas. The major weaknesses of this study include 1) lack of control group; 2) small sample size. Theoretically, most submission will be rejected for interventional study with no control group for comparison. Please if possible provide justification why this study design was performed. The following are more specific comments for your considerations:

Thank you very much for your comment, undoubtedly it was also one of the biggest problems we encountered when facing this study. I will try to justify why we decided to carry it out in this way without a control group.

Indeed, the design with a control group and a larger sample was an important issue for us, but both issues were interrelated due to the possibility of accessing the sample. It was important to us that the sample be homogeneous as much as possible, and the dance academies we had access to did not have a large number of semi-professional or professional dancers who met a minimum weekly practice load and did not have any injuries that could affect the results of the study. We wanted to work with a sample with some experience despite the fact that the largest number of dance practitioners are at much younger ages, but which have much less experience and in addition, due to the variability of development in those ages the sample would have been too heterogeneous and therefore the results, in our view, more subjective.  This led us to have a more homogeneous but much smaller sample.

This sample size led us to decide not to further reduce the size of the experimental group by dividing it into two groups in order to have a control group. In addition, the evidence on the benefits of strength exercise on the health of the entire population was also a determining factor in this decision, so that given the strong potential of strength work, it did not seem entirely ethical that a group of dancers could not benefit from it.

Because of this, it was decided to carry out a pre-posttest study which, despite its limitations, is used in research when a new topic is presented for such a particular population as dancers, to whom, due to tradition and time, this type of physical work is not usually included in their routines.

We have modified the title of the paper to indicate that it is a pilot study, to make it clearer that it is an initial study and that it is intended to lay the groundwork for future studies.

Introduction

Line 47 - 54: not sure if you can find a bit evidence that, to be honest, many pre-professional, semi-professional and professional dancers are required to practice with a very tight schedule. Additional time and efforts on strength training are not their priority........I am thinking if this point is also useful that the training equipment and program must be therefore handy, versatile and convenient to adopt

Thank you very much for your comment, we have included some references that allude to this problem of schedule management. We have also included the need to make practical, versatile and comfortable to adopt programs to facilitate the inclusion of physical programs that help improve physical abilities and injury prevention in dancers.

Line 81 - 84 a bit strange to mention "led by a professional in physical activity and sports"......"design effective and engaging programs...."not sure what is the implication if you study is not to investigate the exercise instruction skill things

We have reworded the paragraph to correct what it states.

Line 55-67: Kettlebell may be more capable in delivering ballistic movement, the release of power and follow-through................just for your consideration if you find this also useful to be added in somewhere

Thank you for your appreciation, it has been included in the introduction.

Line 74-80: I am thinking if this paragraph can be better relocated and also the line 75-76, it is not quite matching the key content of this paragraph

Thank you very much for your suggestion. We have relocated the paragraph to line 60. We have also reorganized the content, changing lines 75-76 to line 58.

Methods:

Line 100 - 7 hours per week only?

This was the minimum time that was established to be included, as they were semi-professionals and all had another occupation. The range of hours ranged from 7 to 18 hours with an average of 10 hours per week.

Figure 1 - ISKF at 90 degree? but the picture is not 90. Also I wonder if the researcher in the picture can hold the leg securely for measuring the powerful hip flexors and knee extensors without a good body position / mechanics

It is indeed an error in the text, and we have already corrected it.

We have modified the figures so that the measurement technique can be better observed, since the previous ones had prioritized the location of the instrument and did not clearly reflect the complete measurement technique.

Any verbal encouragement for isometric test?

The selected protocols did not indicate any verbal stimuli, so no specific words were used to motivate the participants.

Line 139 - what is the meaning of choosing the maximum repetition?

We decided to collect the maximum repetition since we were interested in reaching the maximum force generated. The positions of the protocols are very stable and do not produce large adaptations, however, a lower force generation can occur due to the lack of motivation of the participant. Therefore, the maximum force recorded for the three repetitions was taken.

Line 156 - any warm up required such as 11+ Dance or simple dynamic warm up?

A simple dynamic warm-up was performed. This information has been included in the text (line 165)

Line 188-189 - Why Pearson's correlation used....what is the purpose to identify the association?

We wanted to study whether any anthropometric variable was conditioning isometric strength or jumping capacity, or whether isometric strength was related to jumping capacity.  However, we understand that it is not a consistent test for what we wanted to do and we have modified the analysis. To analyze the influence of the different anthropometric variables and the isometric strength variables on jumping capacity, we performed a generalized linear mixed model (GLMM) taking into account the time of measurement (pretest and posttest) and controlling for interindividual variability. In this way it is possible to observe, without losing the characteristics of the design, the explanatory capacity of the jump (both in parallel and in dehors) of the rest of the variables.

Results:

Table 1 is not presented in a robust manner. It should have Pretest (mean ± SD), Posttest (mean ±SD), Differences (Post-Pre), p-value, Cohen's d value in one table. Also Isometric Strength as a section --> Hip flexors as a sub-section then L and R as two rows......now the left hand side so many words and difficult to read

The table was re-created with the proposed suggestions

Line 208-228: I don't see any big value of reporting all the correlation. If you want to show any change of isolated muscle strength affecting the jumping performance, please consider using a multiple linear regression OR plot a heatmap with corrplot with different colors to help visualize their relationship. Now all these findings are in the long text with poor readability.

A generalized linear mixed model has been performed to replace the linear correlation, and the text has been reformulated accordingly.

Discussion:

Line 230-241: this paragraph can be more precise. It is mainly to highlight your findings or research purpose but now a bit long and clumsy

The paragraph has been reworded to simplify the text.

Line 244: I don't think you can confirm this if you have no control group in your study

We have reworded the paragraph to read as follows:

“A previous study demonstrated that improvements in jump ability in both positions could be achieved with a 5-month training program [14]. In this study, similar results were achieved in just three months. Thus, it seems that the "Simple and Sinister" program carried out in only two weekly sessions over a period of three months, is already showing improvements in the dancers' jumping ability in both parallel and dehors positions.”

Line 251: where is the Borg RPE results? Can you plot a line chart showing the trend of 12 weeks and also the KB weight used as the barchart? Or if this chart is not the best, please if possible also show the result of this part. You mentioned about progressive but we can't fully recognize how progressive load to be implemented in your study

The results of the perception of effort measured by the Borg scale and the progression of the load of each participant have been included in a new figure. Likewise, we have proceeded to discuss the results in the discussion.

Line 258-267: may consider citing some previous studies regarding the required load and dosage to improve the maximum isometric strength. And now your program due to the light load in nature, it may not reach the threshold to induce sufficient intramuscular coordination

This aspect has been incorporated into the discussion, providing references on the dose of isometric force that induces improvements.

Line 268-272: Please CAREFULLY justify why no control group is used in your study. It is the biggest limitation even more severe than the small sample size. Maybe it is a pilot study using such a novel concept in dancers? Maybe the number of semi-professional dancers or professional dancers are always lacking? Anyway, you still need to point this out and also suggest future study to include the control group or even any other interesting group such as fixed load group such that Control vs Progressive vs Fixed ?

Thank you very much for your comment. We have reformulated the limitations section making clear the absence of the control group and justifying it as mentioned above. We have also reflected the need for future research to ad

Reviewer 3 Report

Comments and Suggestions for Authors

Introduction

The description of the need for strength and power training, and how it has historically been avoided, is well done.

The argument for modifying what Grigoletto et al. did is well reasoned.

Methods

Could you please provide a little more detail about the participants, specifically how long they have been doing ballet at a high level and what setting they were in (a professional company, a teaching academy, etc)?

Please describe where they were in their training/performance cycle when this study took place. Think of this as parallel to what may be reported in a team sport: pre-season, in season, etc

lines 128-129: the text describes that the position for the knee flexor test was at 90 degrees of flexion, but in figure 1, the picture for this test shows the person at 0 degrees of flexion (i.e. knee straight)

lines 144-151: Were there any warm up trials or familiarization?

lines 188-190: Make it clear that correlations were only time within each time point; how this reads right now, it makes it seem like correlations between pre and post measures were conducted. For example, if pre-weight correlated with post-jump. After reading the results section, it's apparent that a set of correlations was done for variables at pre-intervention, and another set at post-intervention

I don't see anywhere that describes the timing of the testing relative to the intervention. This may be especially important at post test--if they had trained just a day or two prior to the post-test, they may have still been fatigued or sore, which would have decreased performance and possibly contributed to the finding that few things correlated at post-test

Results

lines 219-221: are you missing the correlation between left hip flexor strength and dehors jump, or was that relationship not significant?

lines 225-228: nothing for you to address here, really, but that is unexpected that most of the muscle strength variables no longer correlated to jump performance.

Discussion

lines 263-267: You have conflated lowering body fat with increasing muscle mass--they are not the same, and don't always happen together.

lines 270-272: You write here that future research should try to improve isometric strength with the goal of improving jump performance, but in your results and in the study you cited in lines 260-261, isometric strength doesn't seem to significantly influence vertical jump. So, why are you making this recommendation?

References

Reference #14, Grigoletto et al., 2020: the doi hyperlink takes you to a 404 webpage not found error; please check the hyperlink for accuracy

Author Response

POINT-BY-POINT FOR THE REVIEWERS

The authors would like to thank all the reviewers for their comments and suggestions, and for their invaluable help in substantially improving our work. We will now proceed to answer each of the questions requested by the reviewers. 

Reviewer 3:

Dear reviewer, we are very grateful for your comments and we will try to resolve and respond to each of them. In the text of the manuscript we will provide the changes highlighted in red. 

Comments to the author

Introduction

The description of the need for strength and power training, and how it has historically been avoided, is well done.

The argument for modifying what Grigoletto et al. did is well reasoned.

Thank you very much for your comments

Methods

Could you please provide a little more detail about the participants, specifically how long they have been doing ballet at a high level and what setting they were in (a professional company, a teaching academy, etc)?

We have expanded the information on sample characteristics in the participants section of the methods section.

Please describe where they were in their training/performance cycle when this study took place. Think of this as parallel to what may be reported in a team sport: pre-season, in season, etc

This information has been included in the procedures section in the methods section.

lines 128-129: the text describes that the position for the knee flexor test was at 90 degrees of flexion, but in figure 1, the picture for this test shows the person at 0 degrees of flexion (i.e. knee straight)

The error has been detected and corrected in the text. We have also redone the images to clarify the position and protocol.

lines 144-151: Were there any warm up trials or familiarization?

In the intervention protocol section, it has been detailed that a simple dynamic warm-up was performed.

lines 188-190: Make it clear that correlations were only time within each time point; how this reads right now, it makes it seem like correlations between pre and post measures were conducted. For example, if pre-weight correlated with post-jump. After reading the results section, it's apparent that a set of correlations was done for variables at pre-intervention, and another set at post-intervention

Due to the reviewers' comments, the correlation analysis was replaced by a generalized linear mixed analysis. Therefore, this section has been completely remodeled.

I don't see anywhere that describes the timing of the testing relative to the intervention. This may be especially important at post test--if they had trained just a day or two prior to the post-test, they may have still been fatigued or sore, which would have decreased performance and possibly contributed to the finding that few things correlated at post-test

Thank you very much for your appreciation. The post-test was performed 24 hours after the last intervention session. It could be that the haste of the measurement could condition the results of the post-test, however in the last session, the perception of effort was 4-5 out of 10 on the Borg scale, so it does not seem that fatigue could condition it. This information has been included in the procedures section of the material and methods section.

Results

lines 219-221: are you missing the correlation between left hip flexor strength and dehors jump, or was that relationship not significant?

The correlation study has been replaced by a generalized linear mixed analysis as suggested by the reviewers.

lines 225-228: nothing for you to address here, really, but that is unexpected that most of the muscle strength variables no longer correlated to jump performance.

We agree with you. We detected problems in the design of the statistical analysis that could be affecting the correlation results. As the reviewers rightly pointed out, we proceeded to perform a generalized linear mixed analysis that controlled for the time of measurement (pretest and posttest) and accounted for interindividual variability.

Discussion

lines 263-267: You have conflated lowering body fat with increasing muscle mass--they are not the same, and don't always happen together.

Thank you very much for your clarification. It is indeed an error. We have reformulated the discussion in this aspect correcting the errors detected.

lines 270-272: You write here that future research should try to improve isometric strength with the goal of improving jump performance, but in your results and in the study you cited in lines 260-261, isometric strength doesn't seem to significantly influence vertical jump. So, why are you making this recommendation?

We understand your concern. We have reformulated future lines of research for clarification.

Regarding your comments, in our study we observed significant relationships between isometric strength and jumping ability in external rotation position. Some of them show that an improvement in isometric strength, such as in hip external rotators or knee flexors, increases the jumping capacity, while this improvement is also related to a decrease in the strength of hip or knee extensors

Therefore, we believe that it would be interesting for future studies to further explore this issue with programs that significantly improve isometric strength.

References

Reference #14, Grigoletto et al., 2020: the doi hyperlink takes you to a 404 webpage not found error; please check the hyperlink for accuracy

Thank you for your comment. However it is the doi indicated in the journal in which the article is published. It is a matter beyond our control. I offer you anyway the link where you can access the article.

https://pmc.ncbi.nlm.nih.gov/articles/PMC7706648/

Round 2

Reviewer 1 Report

Comments and Suggestions for Authors

I want to thank the authors for their revisions. While the manuscript has improved, I still have concerns on the reliability of the isometric testing.

Did you use the maximal value from the test or the average of the three? Please provide your reliability statistics for the testing. Both ICC and CV values would add tremendous value to trusting the data that is reported.

While I appreciate the inclusion of the citations to the protocol and mentioning that one investigator performed all testing, there is still potential for reliability issues. I think that also providing the individual change for each subject would be useful as well.

I do not see how the statistics performed in anyway match the hypothesis stated. You state that the program would change strength and jump performance. So why not leave it as a dependent samples t-test and move on. The regression analysis does nothing to support the hypothesis in my opinion.

Author Response

The authors would like to thank all the reviewers for their comments and suggestions, and for their invaluable help in substantially improving our work. We will now proceed to answer each of the questions requested by the reviewers. 

Reviewer 1:

Dear reviewer, we are very grateful for your comments and we will try to resolve and respond to each of them. In the text of the document we will provide the changes highlighted in red. 

Comments to the author

I want to thank the authors for their revisions. While the manuscript has improved, I still have concerns on the reliability of the isometric testing.

Did you use the maximal value from the test or the average of the three? Please provide your reliability statistics for the testing. Both ICC and CV values would add tremendous value to trusting the data that is reported. While I appreciate the inclusion of the citations to the protocol and mentioning that one investigator performed all testing, there is still potential for reliability issues. I think that also providing the individual change for each subject would be useful as well.

We appreciate the reviewer’s concern regarding the reliability of the isometric testing. We decided to collect the maximum repetition since we were interested in reaching the maximum force generated. The positions of the protocols are very stable and do not produce large adaptations, however, a lower force generation can occur due to the lack of motivation of the participant. Therefore, the maximum force recorded for the three repetitions was taken. However, we proceed to make the calculations you request and include them in the manuscript.

I do not see how the statistics performed in anyway match the hypothesis stated. You state that the program would change strength and jump performance. So why not leave it as a dependent samples t-test and move on. The regression analysis does nothing to support the hypothesis in my opinion.

Thank you very much for your feedback. However, other reviewers found this analysis relevant and interesting. Therefore, we proceeded to perform it. We understand that it may help to reflect on the possible interactions or not of the variables recorded with the jump. We therefore decided to keep it in the paper.

Reviewer 2 Report

Comments and Suggestions for Authors

Thanks for the revision.

As said, most studies without control groups and sufficient power/sample size will be rejected. However, the subject recruitment, or sampling issues you are facing are the same as what I observed in the dancer academies/institutes (including my research partners) in some other European or Asian regions.

Therefore, if the study can give insight, inspiration, and new study direction to the industry, it should be considered for publication at this stage. Your justifications for no control group and a small sample size are very reasonable.

Very few minor amendments I want to highlight:

Line 145: "choosing the maximum repetition"........I am thinking if it should be "the repetition of maximum force output was selected for further analysis"......Now it sounds like you are choosing high repetition rather than high strength

Line 245: It should be Figure 6

Line 228 and 229: if the r=0.41 and r=0.48 are not important, I think the Cohen's d values are already good enough to show the magnitude. Now these two r values are a bit confusing and please consider removing these r values

Line 259: Figure 6 caption.......please state clearly for this figure....the Coefficient estimations of what? Also if possible (correct me if I am wrong), add one more sentence to supplement "The line width shows the magnitude of the coefficient estimations"

Line 313-314: "This may have been the factor that prevented us from obtaining significant improvements in our dancers," --> "This may be the key factor explaining the insignificant isometric strength improvement shown in our study, ...."

One more minor thing, can you report the ICC values of your parallel jump, dehors jump, and isometric strength? Since you have so many isometric strength tests, I think you may report the range such as all isometric strength tests showing good to excellent (0.xx to 0.xx) test-retest reliability in terms of the intraclass correlation coefficient (ICC). Moreover, since you showed significant improvement in parallel and Dehors jump, I think for these two tests, you need to base on ICC values to further calculate the Standard Error of Measurement (SEM) such that readers can better confirm if the observable improvement mainly were from the true improvement of dancers rather than measurement errors.

Line 327: "coordination and efficient activation"...........what about... "neuromuscular control and muscle synergies,....." I think coordination and activation are too generic and not neural specific enough

Line 328: ")", --> .

Line 334: as you have made some speculations/guesses based on your observations, I think you may also want to say "Further studies in this regard are warranted to confirm the relationship between these anthropometric variables and dehors jumping performance"?......then you can delete line 344 to 346 from "As well as ......thigh lengths"

Line 358-361: This part is not very well written. I think the topic of your study is more focusing the low-load KB program on strength and jump performance but NOT the underlying predictive factors or the relationship (these are just side-dish). You may consider rewriting it as "Moreover, the low load strength training program using kettlebells did not significantly increase the muscle mass of our dancers. Therefore, dance teachers and dancers should consider adopting this training method to enhance the dance performance without the significant observable morphological changes (e.g. bulky muscle mass)." ......by emphasizing this, your conclusion can be more impactful and meaningful to many dancers and teachers who may have hesitation on strength training!

Comments on the Quality of English Language

Kind of readable....of course, always can be better but the most confusing parts I have already provided examples for the authors' consideration

Author Response

Dear reviewer, we are very grateful for your comments and we will try to resolve and respond to each of them. In the manuscript we will provide the changes highlighted in red.

 Comments to the author

Thanks for the revision.

As said, most studies without control groups and sufficient power/sample size will be rejected. However, the subject recruitment, or sampling issues you are facing are the same as what I observed in the dancer academies/institutes (including my research partners) in some other European or Asian regions.

Therefore, if the study can give insight, inspiration, and new study direction to the industry, it should be considered for publication at this stage. Your justifications for no control group and a small sample size are very reasonable:

Thank you very much for your comment.

Very few minor amendments I want to highlight:

Line 145: "choosing the maximum repetition"........I am thinking if it should be "the repetition of maximum force output was selected for further analysis"......Now it sounds like you are choosing high repetition rather than high strength

Thank you very much for your suggestion, it is certainly much better understood.

Line 245: It should be Figure 6

Thank you very much, it was indeed a mistake that we have corrected.

Line 228 and 229: if the r=0.41 and r=0.48 are not important, I think the Cohen's d values are already good enough to show the magnitude. Now these two r values are a bit confusing and please consider removing these r values

We have eliminated the r values to avoid confusion.

Line 259: Figure 6 caption.......please state clearly for this figure....the Coefficient estimations of what? Also if possible (correct me if I am wrong), add one more sentence to supplement "The line width shows the magnitude of the coefficient estimations"

We have modified the title of the figure to include the information indicated by the reviewer.

Line 313-314: "This may have been the factor that prevented us from obtaining significant improvements in our dancers," --> "This may be the key factor explaining the insignificant isometric strength improvement shown in our study, ...."

The suggested changes have been made

One more minor thing, can you report the ICC values of your parallel jump, dehors jump, and isometric strength? Since you have so many isometric strength tests, I think you may report the range such as all isometric strength tests showing good to excellent (0.xx to 0.xx) test-retest reliability in terms of the intraclass correlation coefficient (ICC). Moreover, since you showed significant improvement in parallel and Dehors jump, I think for these two tests, you need to base on ICC values to further calculate the Standard Error of Measurement (SEM) such that readers can better confirm if the observable improvement mainly were from the true improvement of dancers rather than measurement errors.

The recommended data has been included

Line 327: "coordination and efficient activation"...........what about... "neuromuscular control and muscle synergies,....." I think coordination and activation are too generic and not neural specific enough

Thank you very much for your suggestion. The text has been modified

Line 328: ")", --> .

Done

Line 334: as you have made some speculations/guesses based on your observations, I think you may also want to say "Further studies in this regard are warranted to confirm the relationship between these anthropometric variables and dehors jumping performance"?......then you can delete line 344 to 346 from "As well as ......thigh lengths"

Thank you very much for your suggestion, it is certainly a good observation. The text has been modified

Line 358-361: This part is not very well written. I think the topic of your study is more focusing the low-load KB program on strength and jump performance but NOT the underlying predictive factors or the relationship (these are just side-dish). You may consider rewriting it as "Moreover, the low load strength training program using kettlebells did not significantly increase the muscle mass of our dancers. Therefore, dance teachers and dancers should consider adopting this training method to enhance the dance performance without the significant observable morphological changes (e.g. bulky muscle mass)." ......by emphasizing this, your conclusion can be more impactful and meaningful to many dancers and teachers who may have hesitation on strength training!

Thank you very much for your comment. It certainly improves substantially the conclusions section of our study.

Round 3

Reviewer 1 Report

Comments and Suggestions for Authors

Thank you for your resubmission. While I still disagree with the statistical analysis that was performed to answer the stated hypothesis, you have made the other adjustments that have been requested.